# A dynamic population of prophase CENP-C is required for meiotic chromosome segregation

**Jessica E. Fellmeth, Janet K. Jang, Manisha Persaud, Hannah Sturm, Neha Changela, Aashka Parikh, Kim S. McKim***

Waksman Institute and Department of Genetics, Rutgers, the State University of New Jersey, Piscataway, New Jersey, United States of America

\* mckim@waksman.rutgers.edu

## Abstract

The centromere is an epigenetic mark that is a loading site for the kinetochore during meiosis and mitosis. This mark is characterized by the H3 variant CENP-A, known as CID in *Drosophila*. In *Drosophila*, CENP-C is critical for maintaining CID at the centromeres and directly recruits outer kinetochore proteins after nuclear envelope break down. These two functions, however, happen at different times in the cell cycle. Furthermore, in *Drosophila* and many other metazoan oocytes, centromere maintenance and kinetochore assembly are separated by an extended prophase. We have investigated the dynamics of function of CENP-C during the extended meiotic prophase of *Drosophila* oocytes and found that maintaining high levels of CENP-C for metaphase I requires expression during prophase. In contrast, CID is relatively stable and does not need to be expressed during prophase to remain at high levels in metaphase I of meiosis. Expression of CID during prophase can even be deleterious, causing ectopic localization to non-centromeric chromatin, abnormal meiosis and sterility. CENP-C prophase loading is required for multiple meiotic functions. In early meiotic prophase, CENP-C loading is required for sister centromere cohesion and centromere clustering. In late meiotic prophase, CENP-C loading is required to recruit kinetochore proteins. CENP-C is one of the few proteins identified in which expression during prophase is required for meiotic chromosome segregation. An implication of these results is that the failure to maintain recruitment of CENP-C during the extended prophase in oocytes would result in chromosome segregation errors in oocytes.

## Author summary

Meiosis in oocytes of diverse organisms, including humans and *Drosophila*, is characterized by a long prophase pause and a cell cycle arrest in meiosis I or meiosis II. These pauses could be a challenge for centromeres, whose components are replenished during G1, and then must remain with the chromosomes until the meiotic divisions. We have investigated the stability, prophase dynamics and function of centromere protein CENP-C. We show that CENP-C expression and loading onto centromeres during prophase is

**Data Availability Statement:** All relevant data are within the manuscript and its Supporting Information files.

**Funding:** J.E.F was supported by a NIH IRACDA post-doctoral Fellowship (1K12 GM093854). This work was supported by NIH grant GM101955 to K. S.M. The funders had no role in study design, data collection and analysis, decision to publish, or preparation of the manuscript.

**Competing interests:** The authors have declared that no competing interests exist.

required for multiple meiotic functions. In contrast, centromere partner CID loaded prior to prophase is maintained until meiotic metaphase, whereas CID chaperone CAL1 is removed from the centromeres. Furthermore, expression of CID during prophase can be deleterious and result in ectopic kinetochore formation. CENP-C loading in prophase is required for sister centromere cohesion and kinetochore assembly. Our results provide the first description of CENP-C dynamics during meiosis and show that prophase expression is required for oocyte spindle assembly and function. CENP-C is among a small number of proteins that are required for the meiotic divisions but are loaded prior to prometaphase. Failure to maintain these proteins during the long prophase of oocyte meiosis may contribute to the increased aneuploidy associated with advanced maternal age.

## Introduction

Errors in meiosis lead to aneuploidy. Aneuploidy occurs more frequently in human oocytes than in sperm, and increases dramatically as women age past 35 years [1,2]. This phenomenon is linked with the advanced age of the oocytes themselves at the time of fertilization. Oocytes are formed prior to birth and arrest in early prophase of meiosis I until ovulation. Centromere localized proteins are amongst those that are found on the chromosomes throughout prophase and therefore could contribute to the mechanisms of aneuploidy related to advanced maternal age.

Failure to maintain the centromere after DNA replication or failure to recruit an effective kinetochore during M-phase can result in chromosome segregation defects and aneuploidy. The chromatin of the centromere is epigenetically marked by the H3 variant CENP-A, known as CID in *Drosophila*, rather than a specific sequence [3–5]. CENP-A interacts with the "constitutive centromere associated network" (CCAN) complex in a wide variety of organisms such as humans and yeast [6]. In many vertebrate cells, two pathways defined by the components CENP-T and CENP-C provide the link between the centromere and outer kinetochore [7]. In some cases, one of these pathways has been lost during evolution. *C. elegans* and *Drosophila* are examples where most or all CCAN proteins have been lost and CENP-C is the only linkage between the inner centromere and outer kinetochore [5].

Centromeres are maintained by loading new CENP-A once per cell cycle, but unlike replication-dependent histones, this does not occur during S-phase. Thus, CENP-A levels are reduced by half during S-phase and return to normal levels once during the cell cycle. The timing of CENP-A loading varies in different species and possibly cell types, but is usually after chromosome segregation in mitosis (M or G1) [3,8–11]. In yeast and mammals, maintenance of centromeric CENP-A depends on CENP-C, the Mis18 complex, and HJURP (holiday junction recognition protein). Mis18 binds to existing CENP-A nucleosomes and is recognized by HJURP to recruit CENP-A in stoichiometric quantities [11–17]. In *Drosophila*, the Mis18 complex and HJURP are absent, but CAL1 takes on a similar role by interacting with both CENP-C and CID [18–21]. CID, CENP-C, and CAL1 rely on each other for maintaining centromeres. CAL1 and CID form a prenucleosomal complex and are recruited to the centromere by CENP-C [11,18,20,22]. The CID-CAL1 interaction promotes centromere maintenance through a feedback loop. Centromeric CAL1 recruits CENP-C, which can then recruit more CID to centromeres [19,23].

In *Drosophila* tissue culture cells and in the syncytial divisions of the embryo, CID is assembled at metaphase and anaphase respectively [9,10]. CID loading was also observed in *Drosophila* sperm at the end of meiosis II, which may be similar to the observed telophase/G1

loading observed in human cell culture and *Xenopus* egg extracts [8,24–26]. The timing of CID loading in germ line cells, however, may differ. CID assembly in the germline stem cells of *Drosophila* ovaries occurs in G2 [27]. Similarly, CID assembly occurs during prophase I of *Drosophila* oogenesis and spermatogenesis [24,28,29].

While CENP-C is required for centromere maintenance, it also has a direct role in kinetochore assembly by recruiting Mis12 in mammals [30–32] and *Drosophila* [33–35]. In many organisms, including *Drosophila* and mammals, the kinetochore is loaded once the nuclear envelope breaks down, and is only present during cell division. In contrast, CENP-C is present throughout the cell cycle and is observed at all stages of meiotic prophase in *Drosophila* oogenesis [36]. In *Drosophila*, no other kinetochore proteins except for Mis12 exhibit this behavior [37]. CENP-C recruits Mis12 [34,35], and they directly interact to form an inner kinetochore complex that is present throughout the cell cycle [31,33,38].

Following the pachytene stage of meiosis, mammalian and *Drosophila* oocytes enter a long pause in prophase. If centromere proteins are loaded once per cell cycle, they would have to be maintained for a long time before kinetochores are assembled and cell division commences [39]. Cohesins, for example, are loaded during S-phase and deteriorate with age, causing aneuploidy in older mothers [40,41]. CENP-A nucleosomes in mouse oocytes are stable and do not need to be maintained during meiotic prophase [39,42]. Furthermore, CENP-C becomes immobilized at metaphase [43]. Thus, it is possible that the CENP-C loaded along with CENP-A in G1 is sufficient for kinetochore assembly. However, the stability of CENP-C during meiotic prophase has not been examined.

In this study, we investigated the meiotic prophase dynamics of CENP-C, and compared to CAL1, and CID in *Drosophila* oocytes. As with CENP-A in mouse oocytes [39], we found that CID is stable throughout *Drosophila* oocyte meiotic prophase, although there may also be addition of new subunits. Over expression of CID during prophase can lead to ectopic kinetochore formation and abnormal chromosome segregation. CAL1 is gradually lost during prophase and is absent in metaphase I oocytes. CENP-C is unique because it is loaded at centromeres using an exchange mechanism during meiotic prophase, and this is required for assembly of the meiotic kinetochore. Thus, CENP-C is loaded independently of centromere maintenance in a process that is required for kinetochore assembly.

## Results

### CENP-C loads onto centromeres during prophase I

The analysis of centromere protein dynamics during prophase is facilitated by the organization of the *Drosophila* ovary. Each of the two *Drosophila* ovaries contains several strings, or ovarioles, of oocytes arranged in developmental order (Fig 1A). At the anterior end of each ovariole is the germarium, which includes mitotically dividing cells (region 1) and early meiotic prophase (regions 2–3). Region 3 (stage 1) oocytes leave the germarium and enter the vitellarium. Stages 1 to 13 last 5 days, during which meiosis remains in prophase while the oocyte grows and matures [44,45]. At the posterior end of the ovariole (stage 13–14), the oocyte enters prometaphase. Meiosis arrests at metaphase I until passage through the oviduct and fertilization occurs.

To determine if CENP-C, CAL1, or CID can assemble on chromosomes during meiotic prophase, we used *UASP* regulated EGFP-tagged transgenes. With these transgenes, expression of *Cenp-C*, *cal1*, and *cid* was induced during oocyte meiotic prophase using one of three *GAL4*-expressing transgenes (Fig 1A). This expression was in addition to endogenous *Cenp-C*, *cal1*, or *cid*, possibly resulting in overexpression. We started with *P{GAL4::VP16-nos.UTR} 0CG6325*[MVD1] (referred to as *MVD1*) because it promotes expression of UAS transgenes

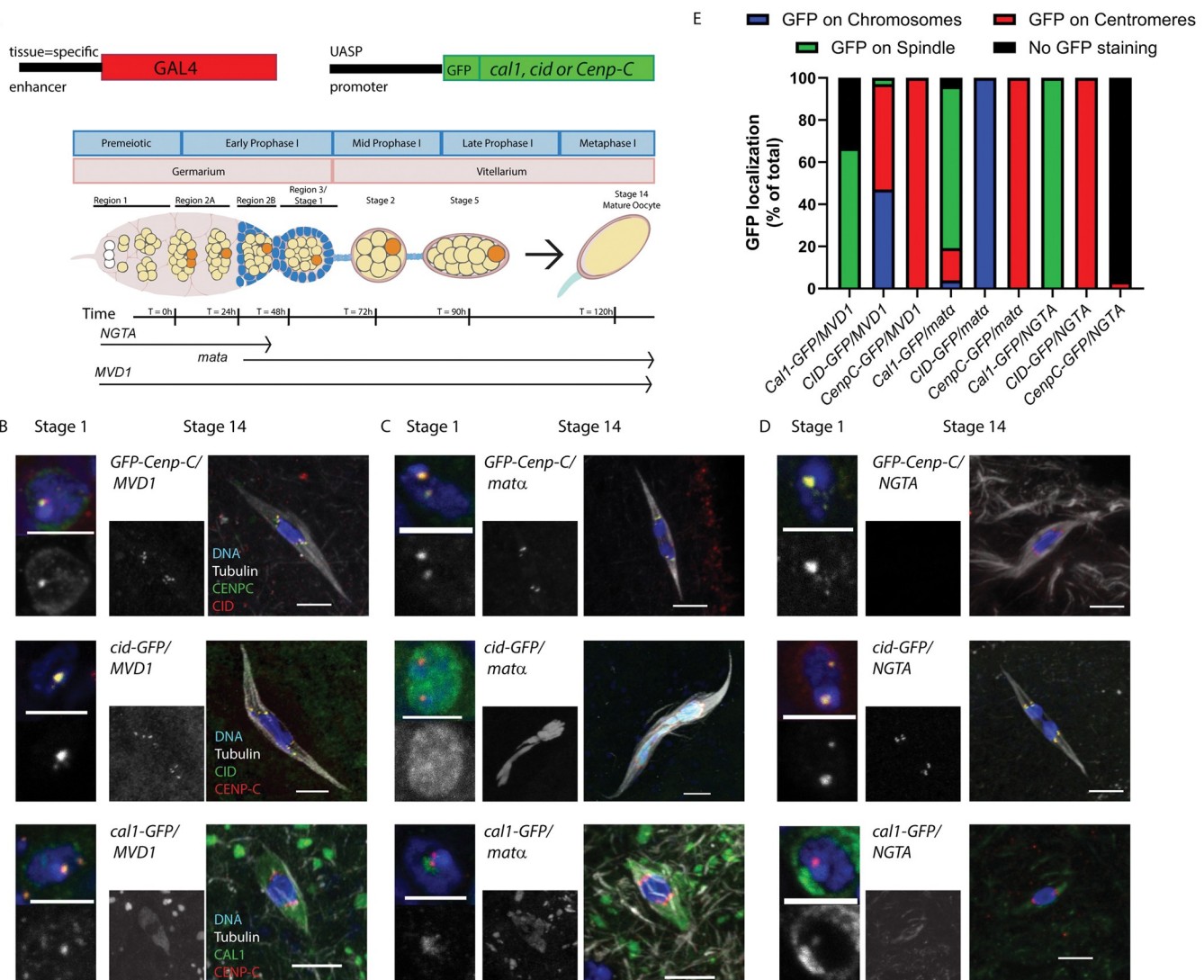

**Fig 1. Loading of centromere proteins during meiotic prophase in oocytes. A)** Oocytes are generated and enter meiotic prophase in the germarium. Time points are relative to the onset of meiosis [44,45]. Region 1 contains the stem cell niche as well as the pre-meiotic cyst cells undergoing mitosis. Prophase I of meiosis begins in region 2A. Region 3 is also known as vitellarium stage 1, and a single oocyte (orange) has been determined. Three versions of the transcriptional activator GAL4 were used to express centromere proteins in a wild-type background. *MVD1* expresses in all the mitotic and meiotic stages of the germline, including the stem cells. *Mata* begins expressing in region 2B or 3 and continues until stage 14, the metaphase I-arrested oocyte [47]. *NGTA* begins expressing in the mitotic germline (region 1) and ends in region 3 or early in the vitellarium (S3 Fig). **B)** Localization of GFP-tagged centromere proteins (green) regulated by *MVD1*. **C)** Localization of GFP-tagged centromere proteins (green) regulated by *mata*. **D)** Localization of GFP-tagged centromere proteins (green) regulated by *NGTA*. In all images, the DNA is blue, centromeres were detected using a CENP-C or CID antibody (red), and the scale bars are 5 μm. **E)** Summary of CID, CENP-C, and CAL1 localization in Stage 14 oocytes (n = 15, 34, 12, 26, 18, 10, 4, 11, 30).

through all stages of the germline, including the pre-meiotic cyst cells of the germline and continuing throughout all stages of oocyte development [46]. Crossing each transgene to *MVD1* resulted in CAL1, CID, and CENP-C centromere localization in region 2a and throughout stages 1 to 5 of oogenesis (Figs 1 and S1).

As expected, CENP-C regulated by *MVD1* (referred to as *GFP-Cenp-C/MVD1* oocytes) was localized to meiotic centromeres in metaphase I of stage 14 oocytes. In contrast, surprising patterns were observed in stage 14 oocytes when CID or CAL1 were regulated by *MVD1* (Fig 1B). In *cid-GFP/MVD1* oocytes, CID was present as centromeric foci in about 50% of oocytes

(Fig 1E). In most of the other oocytes, however, CID was present on the meiotic chromatin. It was enriched in puncta and appeared to recruit CENP-C to these ectopic sites (S2 Fig). Thus, overexpression of CID in prophase can result in non-centromeric chromatin localization. In contrast, CAL1 was absent from the centromeres in *cal1-GFP/ MVD1* stage 14 oocytes (Fig 1B). Consistent with this observation, CAL1 was also absent from stage 14 oocytes when expressed with its own promoter (*gCal1-GFP*, S2 Fig). There was some spindle localization with *MVD1*, which could be due to overexpression because it was not observed when CAL1 was regulated by its own promoter (Figs 1B and S2). Thus, CAL1 appears to be unloaded from meiotic centromeres during prophase.

To test if centromere proteins load during meiotic prophase, we used *P{w[+mC] = matalpha4-GAL-VP16}V37* (referred to as *mata*), which begins expression in late prophase (region 3) and continues through metaphase I in the stage 14 oocyte [47,48]. In *cal1-GFP/ mata* oocytes, a weak GFP signal was frequently observed on the spindle, and rarely at the centromeres (Fig 1B and S2). This is consistent with the observations made in *cal1-GFP/ MVD1* oocytes that CAL1 is absent from metaphase I centromeres. CID induced by *mata* during late prophase I was present on most of the meiotic chromatin in 100% of the oocytes (Fig 1C and 1E). This was similar to the pattern observed in some of the *cid-GFP/ MVD1* oocytes; CID was enriched in some puncta, and also appeared to recruit CENP-C and kinetochore proteins like Spc105R, the *Drosophila* homolog of Knl1, to the meiotic chromatin (S2 Fig). Ectopic CID affected chromatin organization, as individual bivalents could be discerned, unlike wildtype metaphase I oocytes in which the chromosomes are assembled in a single karyosome. These females were also sterile, suggesting that overexpression of CID during prophase is deleterious, resulting in CID localizing to most of the chromatin, recruiting other centromere and kinetochore proteins, and causing changes in chromosome organization.

To test if early prophase loading was sufficient for later stages of meiosis, we used *P{GAL4-nos.NGT}A* (referred to as *NGTA*), which expresses in the germarium, including the mitotic divisions and early meiotic prophase (S3 Fig). Importantly for our study, *NGTA* expression ends in early prophase (stages 3–5). When regulated by *NGTA*, centromeric localization of CID, CAL1, and CENP-C was observed in region 2a and stages 1–5, consistent with results obtained with *MVD1* (Figs 1D and S1). Similar results were observed with a different *Cenp-C* transgene, *HA-Cenp-C* (S4A Fig). These results correspond to the *NGTA* expression pattern (S3 Fig). In stage 14 oocytes, considerably after the zone of *NGTA* expression, CID was localized at the centromeres (Fig 1D). While we can't rule out that *cid* transcripts made early in oogenesis persist into late oogenesis, the most likely interpretation of this data is that CID loaded during or prior to early meiotic prophase is maintained throughout prophase and into metaphase I of stage 14 oocytes. As with *MVD1*, CAL1 was not detected at centromeres in *cal1-GFP/ NGTA* stage 14 oocytes (S1 Fig), confirming that CAL1 is unloaded from centromeres during meiotic prophase. While *GFP-Cenp-C/ NGTA* oocytes had centromeric CENP-C localization in stages 1–5, it was absent in stage 14 oocytes (Fig 1D and 1E). These results suggest that CENP-C is also unloaded from the centromeres during meiotic prophase.

In summary, CENP-C localized to the stage 14 oocyte centromeres in *GFP-Cenp-C/ mata*, but not *GFP-Cenp-C/ NGTA* oocytes (Figs 1C and S4A,S4B). These results show that CENP-C is highly dynamic during meiotic prophase. In support of this conclusion, heat shock was used to induce a pulse of expression during prophase and resulted in localization of CENP-C at all timepoints observed (S4C Fig). These data indicate that CENP-C is recruited to centromeres during meiotic prophase. In contrast, CID expression during or prior to early prophase is sufficient for meiosis I while CAL1 is lost. Thus, meiotic CENP-C dynamics are different than CID and CAL1. CENP-C expression late in prophase is required for its centromere localization in meiosis I.

## Centromeric CENP-C exchanges during prophase I

The results with *NGTA* and *mata* used to express *GFP-Cenp-C* suggest that centromeres unload and load CENP-C during meiotic prophase. To test the relationship between the unloading and loading of CENP-C, females were generated that expressed two forms of CENP-C in addition to the endogenous gene (Fig 2A). The *gGFP-Cenp-C* transgene was under the control of the endogenous promoter, and the *HA-Cenp-C* transgene was under the control of *UAS*, and therefore, *mata*. With these transgenes we could measure if the HA-CENP-C induced in prophase would replace the constitutively expressed GFP-CENP-C present at the centromeres. In addition, we knocked down *gGFP-Cenp-C* expression in prophase by RNAi using shRNA *GL00409*, which is also regulated by the UAS enhancer and therefore, by *mata*. The target sequence of *GL00409* is within the 5' UTR of *Cenp-C*, and therefore, not present in the *HA-Cenp-C* transgene. GFP-CENP-C was expected to be expressed at all stages of the germline but, due to containing the 5' UTR, would be sensitive to RNAi in meiotic prophase. In contrast, HA-CENP-C expression would begin in early prophase and be resistant to RNAi because it lacks the 5'UTR. These genotypes allowed us to observe whether cytoplasmic HA-CENPC was being exchanged with centromeric GFP-CENPC during prophase.

In *gGFP-Cenp-C, HA-Cenp-C/ mata* (no RNAi) oocytes, the GFP and HA variants were both maintained which is consistent with the earlier observations of *GFP-Cenp-C* expressed with *matα* (Fig 2B and 2E). A severe reduction in stage 5 and stage 14 GFP was observed in *gGFP-Cenp-C, HA-Cenp-C, GL00409/ mata* females, consistent with a knockdown of *gGFP-Cenp-C* but not *HA-Cenp-C* (Fig 2C and 2E). In contrast, the decrease in GFP-CENP-C was less severe in the absence of HA-CENP-C expression (*gGFP-Cenp-C, GL00409/ mata* females, Fig 2D and 2E). To explain these observations, we propose that the unloading of centromeric CENP-C depends on the availability of a cytoplasmic replacement. Without the HA replacement, the GFP-CENP-C stayed on the centromeres. In addition, while the GFP-CENP-C levels at the centromeres drop dramatically in the presence of HA-CENP-C and RNAi, a low level of GFP was still observed at centromeres in stage 14 oocytes (Fig 2D). These results suggest that, although most CENP-C is exchanged during prophase, there is a small pool of centromeric CENP-C that is maintained throughout meiosis.

## CENP-C regulates centromere clustering, recombination, and Mis12 loading in early prophase I

These results show that there is a population of CENP-C loaded during meiotic prophase, which we will call the prophase I pool, that is distinct from the population required for the maintenance of CID at the centromeres, which we will call the premeiotic pool. To investigate the function of the *Cenp-C* prophase I pool, we knocked down endogenous expression by RNAi. Using *GL00409* with *MVD1* resulted in *Cenp-C* knockdown throughout the germline, effecting both pools. Females expressing *GL00409* with *MVD1* (*GL00409/MVD1* oocytes) were sterile and agametic and is consistent with previous studies [49,50]. *MVD1*-driven expression in all premeiotic mitotic germ cells and a requirement for CENP-C to load CID during these mitotic divisions would explain the agametic phenotype,

Using *GL00409* with *NGTA* was expected to knockdown *Cenp-C* early in meiosis, possibly effecting only the prophase I pool. *GL00409/NGTA* females were fertile but displayed a high level of X-chromosome nondisjunction (7%) compared to controls (0.25%) (Table 1). A similar result was observed with a different shRNA targeting *Cenp-C* (*HMJ21500*). Because these females produce oocytes, and *NGTA* expression occurs only during early prophase, these results suggest the prophase I pool of CENP-C has a function early in meiotic prophase required for chromosome segregation. To investigate the mechanisms of nondisjunction

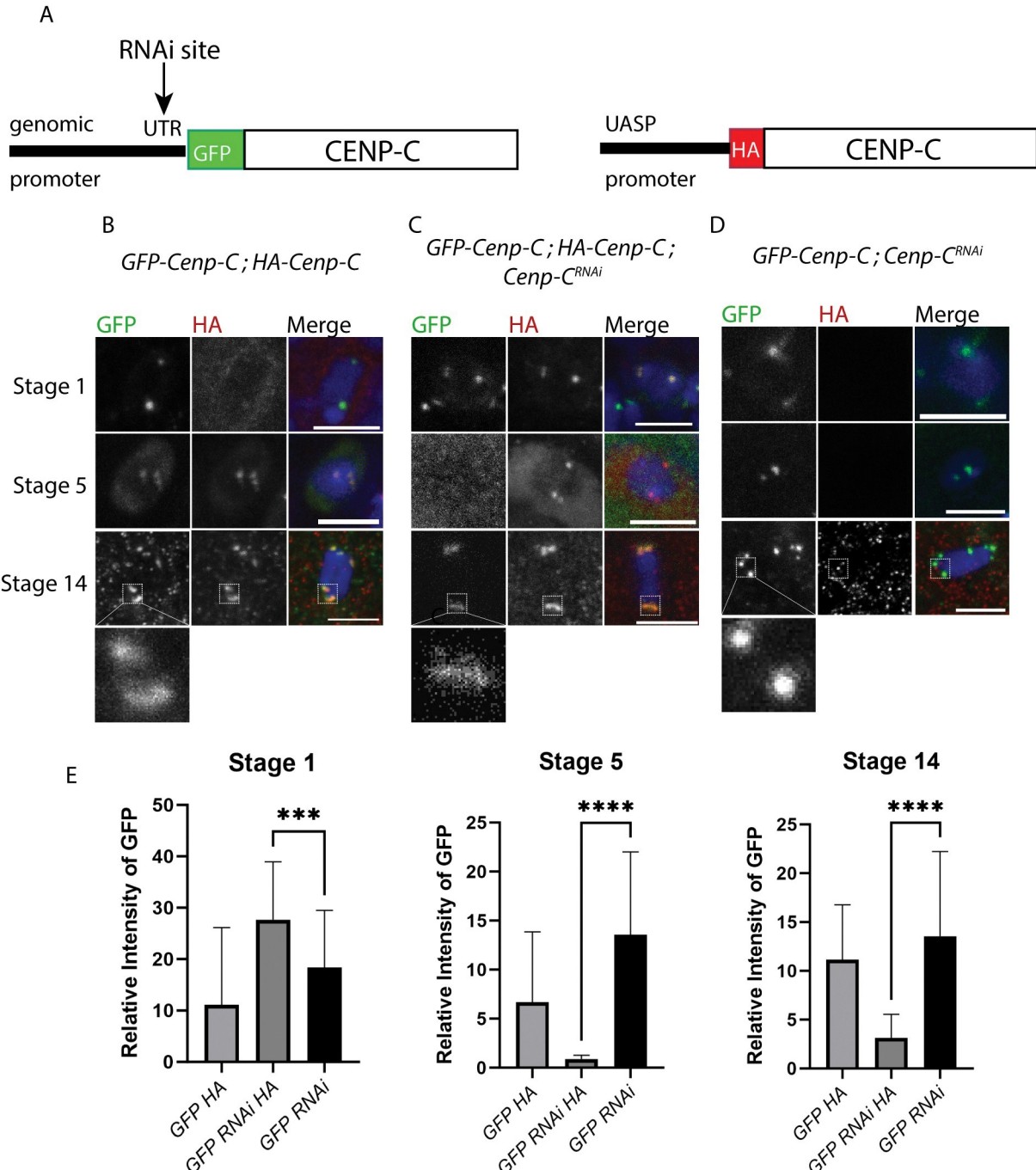

**Fig 2. CENP-C is exchanged at the centromeres during meiotic prophase.** Immunocytology was performed on oocytes ubiquitously expressing GFP-CENP-C (green), plus or minus *GL00409* (*Cenp-C^RNAi*), and HA-CENP-C (red) using the *mata* promoter. All scale bars represent 5 µm and DNA is blue. **A)** *GL00409* recognizes a sequence in the 5' UTR, and, therefore, degrades *GFP-Cenp-C* but not *HA-Cenp-C*. Images show stages 1, 5 and 14 with (**B**) expression of GFP-CENP-C and HA-CENP-C, (**C**) expression of GFP-CENP-C, *Cenp-C^RNAi*, and HA-CENP-C, and (**D**) expression of GFP-CENP-C with *Cenp-C^RNAi*. **E)** The relative intensity of GFP-CENP-C was measured at each stage in each genotype (n = 70, 117, 28, 23; 12, 19, 104, 156; 50) with error bars showing standard deviation. For each genotype, the intensity between stage 1 and stage 14 was compared using an unpaired t-test (***p = 0.001; ****p<0.0001).

**Table 1. Fertility, nondisjunction and crossing over in *Cenp-C* RNAi females.**

| | Fertility | Nondisjunction (NDJ) | | Recombination (% of control) | | | |
|---|---|---|---|---|---|---|---|
| | Offspring per female (# of females) | Homologous Chromosome NDJ (# of offspring) | Sister Chromatid NDJ (# of offspring) | *st-cu (m. u.)* | *cu-e (m.u.)* | *e-ca (m. u.)* | # of progeny |
| Control | 24.3 (105) | 0.25% (1590) | 0.5% (2696) | 4.4 | 2.2 | 29.4 | 2853 |
| *GL00409 / NGTA* | 10.0 (240) | 7.0% (1658) | 2.6% (1164) | 9.4 (226%) | 21.1 (101%) | 22.9 (81%) | 1668 |
| *HMJ21500/ NGTA* | 3.3 (133) | 7.6 (1213) | ND | ND | ND | ND | ND |
| *GL00409 / MVD1* | 0.27 (105) | 0% (54) | - [a] | - | - | - | - |

[a] not enough progeny to accurately measure nondisjunction or crossing over

caused by loss of CENP-C, we characterized processes that occur early in meiotic prophase such as centromere assembly, meiotic recombination, centromere pairing, sister chromatid cohesion, and synaptonemal complex (SC) formation.

The intensities of CENP-C and CID at the centromeres were moderately decreased, when measured using CENP-C and CID antibody staining, in early prophase nuclei of *GL00409/ NGTA* oocytes (Fig 3A–3C). The mild reduction of both centromere proteins suggests that the knockdown of CENP-C only had a mild effect on centromere maintenance. For another measure of CENP-C function in prophase, we examined the localization of Mis12. In *Drosophila* mitotic cells, CENP-C promotes kinetochore assembly by recruiting Mis12 [30,33–35,51]. Mis12-GFP localized to centromeres in control oocytes, starting in region 2a of the germarium (Fig 3D). The intensity of centromeric Mis12-GFP was significantly reduced in *GL00409/ NGTA* females (Fig 3E), indicating that CENP-C promotes recruitment of Mis12 during meiotic prophase. To explain why Mis12 is more significantly reduced than CENP-C, we propose that the recruitment of Mis12 relies on the prophase I pool of CENP-C, which is shown by the amount of CENP-C reduction in Fig 4B. The remaining CENP-C may represent the premeiotic pool that was loaded prior to prophase and may not recruit Mis12. Further experiments are required to investigate this possibility.

Increased frequencies of nondisjunction can be associated with loss of SC and/or a reduction in crossing over (CO) [52]. SC assembly in *GL00409/NGTA* females was normal, indicated by the thread-like appearance of the transverse filament protein C(3)G [53] (S5 Fig). We measured the frequency of crossing over in *GL00409/NGTA* females and observed an increase in centromeric crossovers, but overall, crossing over was not decreased (Table 1). When the centromeres were genetically marked (see Methods, Table 1), we observed an increase in sister chromatid nondisjunction. Thus, meiotic nondisjunction may occur because of a cohesion defect. The effect on proximal crossovers could be secondary and related to the suppressing effect of centromeres [54,55].

A cohesion defect can be observed as an increase in the number of centromere foci [56,57]. The centromeres usually cluster in meiotic prophase, resulting in two or three CID foci. There was an increased number of CID foci in the germaria of *GL00409/NGTA* ovaries, indicating a separation of centromeres (Fig 3F and 3G). We tested if the cohesion defect persisted, and therefore would result in increased CID foci in metaphase I oocytes. The variability in CID foci number was greater in the experimental group, but the mean was not significantly different from the control (Fig 3H and 3I).

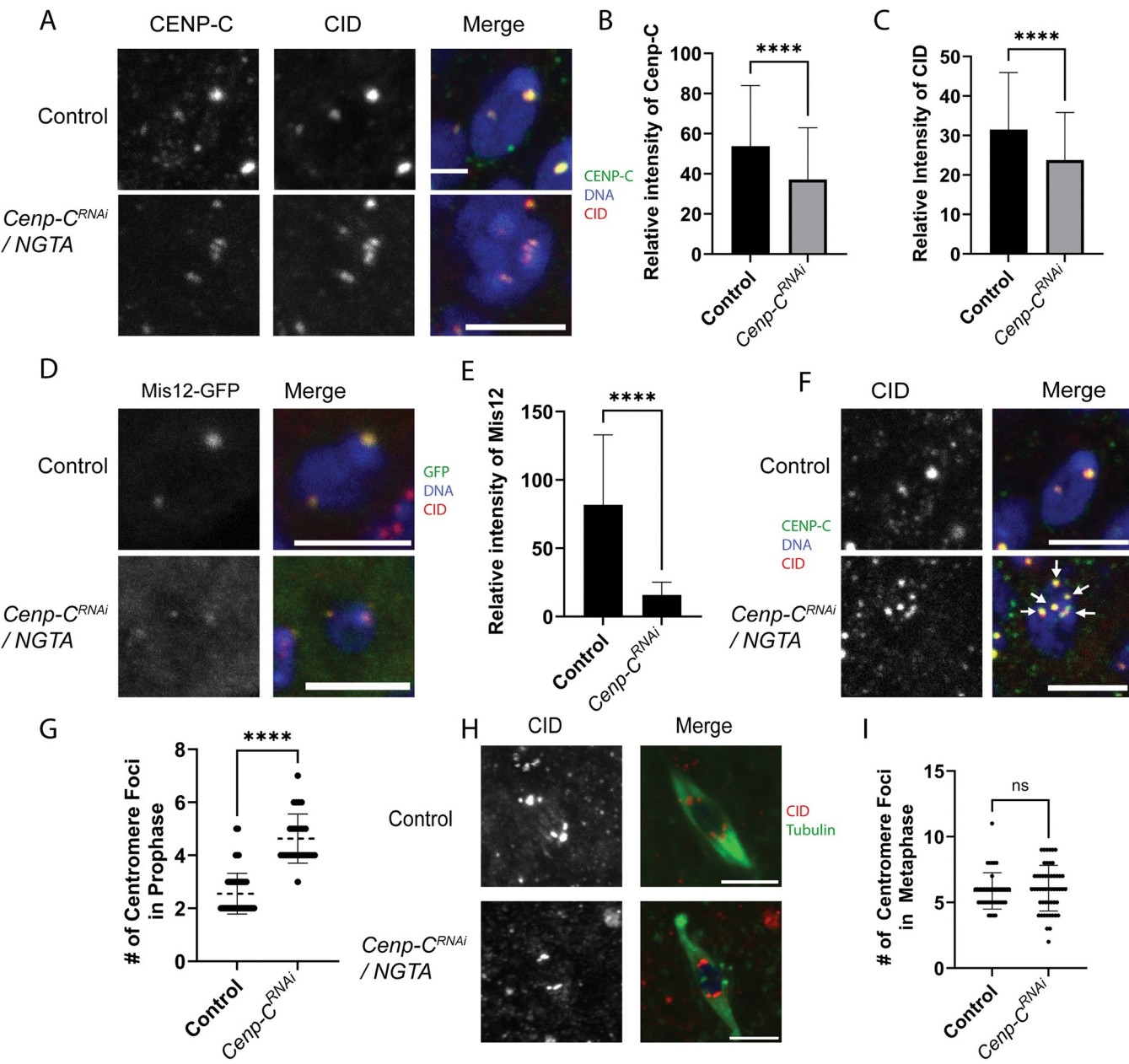

**Fig 3. Early Prophase I defects when CENP-C is depleted using NGTA.** Immunocytology was performed on *GL00409/NGTA* (referred to as *Cenp-C^RNAi^*) oocytes with CID (red), DNA (blue), and scale bars represent 5 μm. **A)** Region 2 oocytes showing CENP-C in green. **B-C)** The intensities of CENP-C and CID in region 2 were measured in *Cenp-C* RNAi and control oocytes (For B, n = 104 and 88; for C, n = 104 and 87). **D)** Region 2 oocytes with Mis12-GFP in green. **E)** Mis12 intensity was measured in region 2 (n = 13 and 30). **F)** CID foci in *Cenp-C* RNAi region 3 oocytes, with CENP-C in green. **G)** Number of centromere foci was measured based on CID foci in contact with the DNA (n = 84 and 27). **H)** Tubulin (green) and CID foci (red) in *Cenp-C* RNAi metaphase I oocytes. **I)** There was no increase in CID foci observed in stage 14 oocytes (n = 40 and 52). Error bars represent standard deviation from the mean. ****p<0.0001.

Because nonhomologous centromeres normally cluster in early prophase, we used chromosome-specific fluorescence in situ hybridization (FISH) probes for the pericentromeric regions of the X and 2^nd chromosomes to examine pairing and clustering of the centromeres (Fig 4A). Clustering defects were defined as nuclei containing 3 or more clearly separated probe foci (of any color) whereas control oocytes had 1–2 foci. As with the experiment to detect CID foci, the FISH foci failed to cluster (Fig 4B). Pairing defects were defined as greater than one FISH

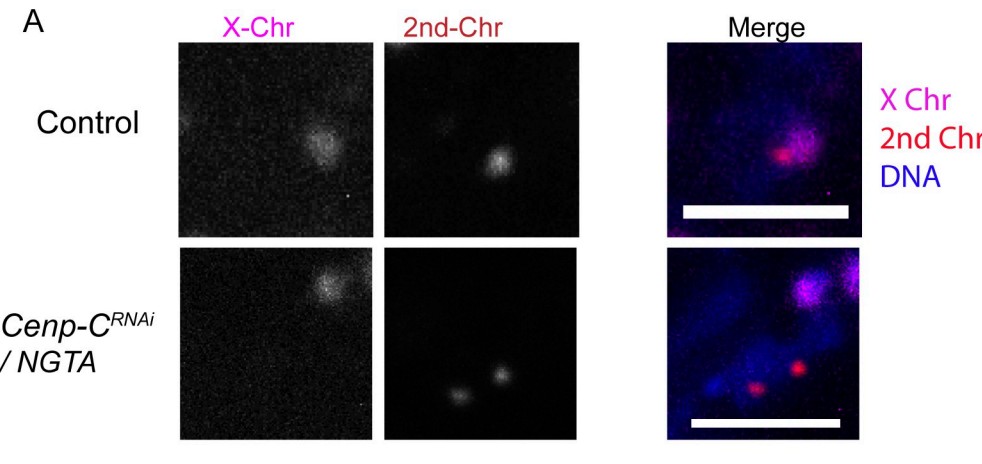

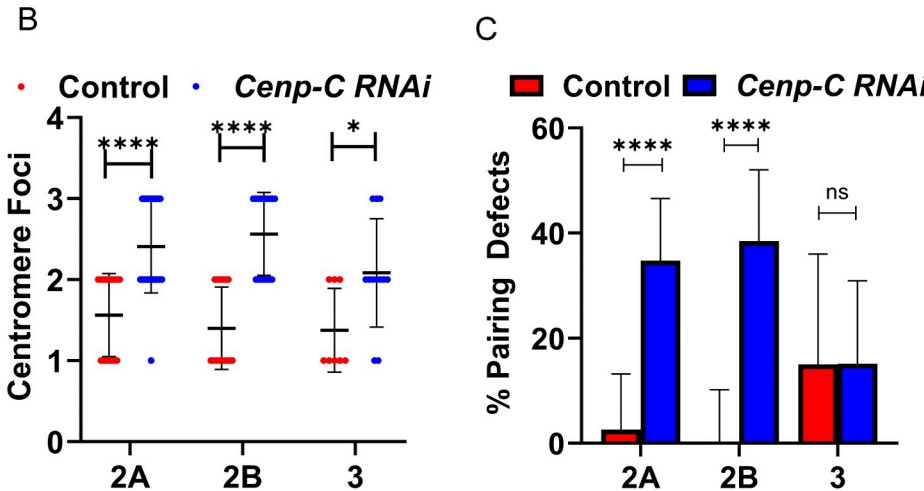

**Fig 4. CENP-C is required for centromere pairing and clustering. A)** FISH probes were used to detect the pericentromeric regions of chromosome 2 (red), and X (magenta) in the germarium of *GL00409/NGTA* (referred to as *Cenp-C^RNAi*) or control oocytes. DNA is in blue. **B)** Clustering defects were defined as nuclei with greater than 2 of any centromere foci (control n = 16, 20, 14; RNAi n = 42, 38, 18). **C)** Pairing defects were defined as oocytes with greater than one focus for a chromosome in a nucleus. Scale bars represent 5 μm and error bars show standard deviation. * = 0.0196>p>0.0238, ****p<0.0001.

focus per probe. While homologous centromeres were efficiently paired in wild-type germarium, pairing was defective in *GL00409/NGTA* oocytes (Fig 4C). These results suggest that there are defects in both centromere pairing and clustering, consistent with a defect in sister centromere cohesion. These results are similar to previous results with a hypomorphic allele of *Cenp-C* [58], but extend them by showing this function depends on the prophase I pool of CENP-C.

## Prophase CENP-C plays a critical role in maintenance of centromeres and kinetochores

To test the function of the prophase I pool of CENP-C, we used *mata* to express *Cenp-C* shRNA (*GL00409/mata*). The *GL00409/mata* females were sterile, although this could be due to defects in early embryo mitosis rather than meiosis. To investigate the effects on meiosis I,

stage 14 *GL00409/mata* oocytes were analyzed for meiotic defects. The intensity of CENP-C at the centromeres was reduced to approximately 25% of the intensity in control oocytes (Fig 5A and 5B). The intensity of CID at the centromeres was mildly reduced in *GL00409/mata* oocytes to approximately 80% of control levels (Fig 5A and 5C). These results suggest that CENP-C might play a role in the stabilization or loading of CID late in meiotic prophase or in metaphase I.

To determine if the loss of CENP-C affects kinetochore assembly, we quantified the intensity of Mis12 and kinetochore protein Spc105R in mature oocytes. Consistent with the reduction in CENP-C levels, we observed approximately 85% reduction in Mis12 levels and a 90% reduction in Spc105R in *GL00409/mata* oocytes (Fig 5D–5G). These results show that the prophase I pool of CENP-C is required to assemble a kinetochore.

Consistent with the loss of Spc105R recruitment, we observed an increase in the distance between centromeres and microtubules (S6 Fig), which has previously been observed when Spc105R is depleted [59]. These results suggest CENP-C recruits Mis12, which then recruits Spc105R, although we can't rule out CENP-C recruits Spc105R directly. We previously showed that loss of Spc105R was associated with loss of sister centromere cohesion [60]. However, we did not observe this phenotype in *GL00409/mata* oocytes (S6 Fig), suggesting sufficient Spc105R remains to protect cohesion.

To determine whether the loss of CENP-C in oocytes affected chromosome segregation at meiosis I, we assayed for bi-orientation of homologous chromosomes using FISH. We observed a dramatic increase in bi-orientation errors (primarily mono-orientation) in *GL00409/mata* oocytes (Fig 6A and 6B), and a decrease in the distance between homologous centromeres indicating a loss of microtubule attachments, compared to the control (Fig 6C). These spindle attachment errors could indicate that CENP-C plays a direct role in error correction, but more likely is due to the requirement of CENP-C for kinetochore assembly [61–63].

## Rescue of the CENP-C phenotype by late prophase expression of a wild-type allele

The defects in *GL00409/mata* females suggest that the prophase I pool of CENP-C expression is required for meiosis and fertility. To directly test if the prophase I pool of CENP-C is functional in meiosis, we determined if the *HA-Cenp-C* transgene could rescue loss of endogenous *Cenp-C*. The *HA-Cenp-C* transgene lacks the 5'UTR that includes the target sequence for *GL00409*, and therefore, is resistant to knockdown by RNAi. When *HA-Cenp-C* was co-expressed with *GL00409* using *mata* in oocytes, we observed localization of HA-CENP-C to the centromeres and restoration of Mis12 and Spc105R localization (Fig 5D–5G). In addition, the biorientation defects in *GL00409/mata* oocytes were rescued by expression of HA-CENP-C (Fig 6). Thus, prophase expression of a CENP-C transgene rescued the defects observed in RNAi oocytes.

Despite the rescue of the meiotic RNAi phenotypes, however, expression of HA-CENP-C using *mata* caused sterility (Table 2). These results suggest overexpression of CENP-C leads to loss of fertility. The females expressing HA-CENP-C had normal metaphase I oocyte spindles, making it likely that the defect causing sterility is after meiosis I, possibly during embryonic mitosis. To circumvent the sterility from *mata*-induced expression of *Cenp-C*, we used *P{w [+mC] = osk-GAL4::VP16}A11* (referred to as *oskGal4*). The expression pattern of *oskGal4* is similar to *matα* but at lower levels [64]. Using *oskGal4* to induce *UASP*-regulated *Cenp-C*, either GFP-tagged or HA-tagged, resulted in fertile females (Table 2).

We then used *oskGal4* to determine if prophase expression of CENP-C could rescue the phenotype of a hypomorphic allele, *Cenp-C^{Z3-4375}* [58]. When *Cenp-C^{Z3-4375}* was heterozygous

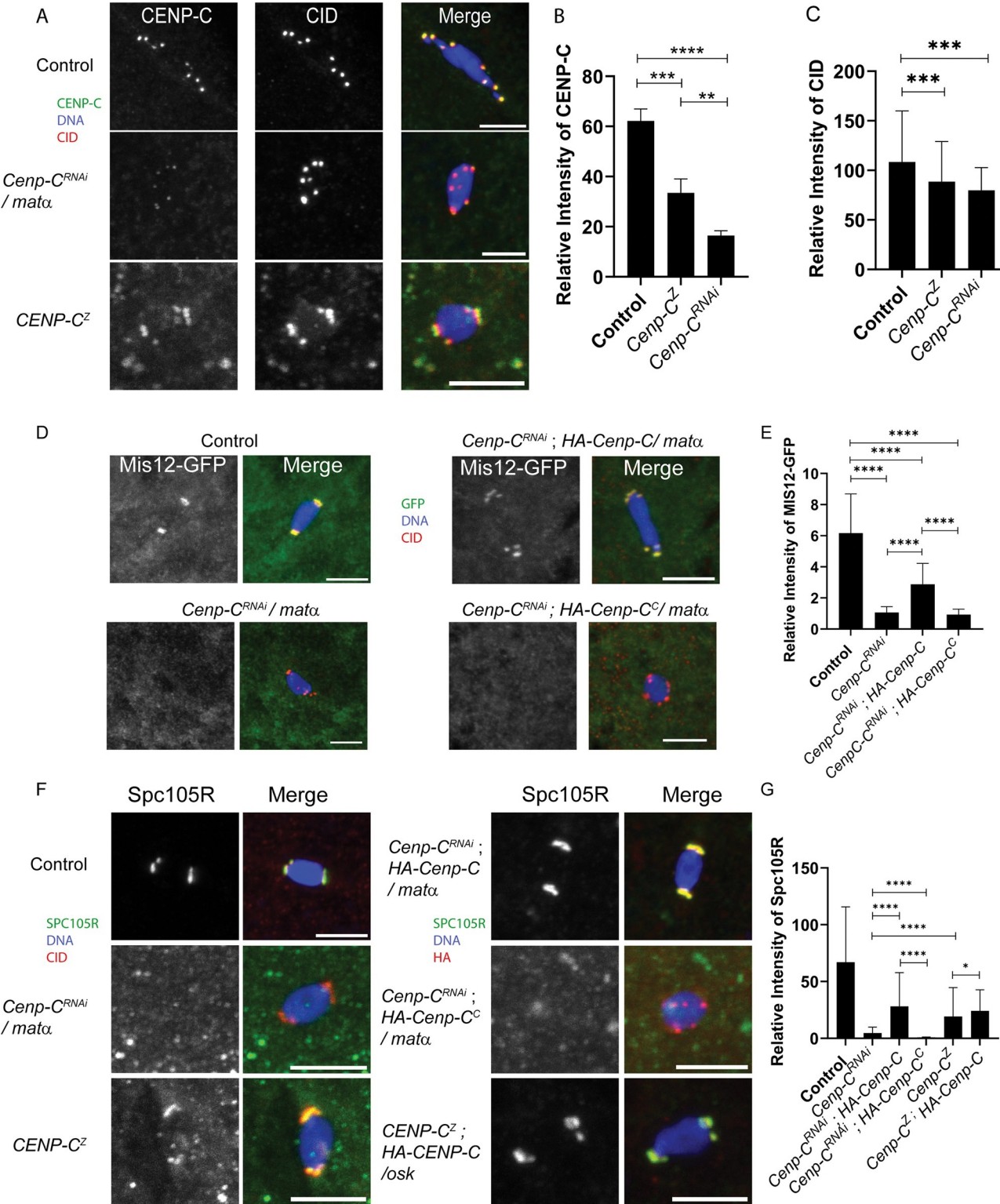

**Fig 5. CENP-C is required for kinetochore assembly and chromosome segregation.** Immunocytology was performed on *GL00409 / matα* stage 14 oocytes (referred to as *Cenp-C^RNAi*) or a *Cenp-C* mutant (*Cenp-C^Z3-4375*/*Cenp-C^IR35* referred to as "*Cenp-C^Z*"). HA-tagged *Cenp-C* transgenes were coexpressed using the *matα* promoter in RNAi experiments or *osk-Gal4* in *Cenp-C^Z* experiments. DNA is in blue and scale bars represent 5 μm in all images. **A)** Oocytes with CENP-C (green) and CID (red). **B)** The intensity of CENP-C was measured relative to background (n = 55, 67, 54). **C)** The intensity of CID was measured relative to background (n = 186, 221, 54). **D)** Oocytes expressing Mis12-GFP (green) and CID (red). **E)** Mis12-GFP localization was measured relative to the background (n = 57, 41, 47, 67). **F)** Oocytes with Spc105R (green) and CID (red). **G)** Intensity of Spc105R was measured relative to background (n = 108, 136, 115, 132, 245, 150). *p = 0.0379, **p = 0.0086, *** = 0.0002>p>0.0001, ****p<0.0001.

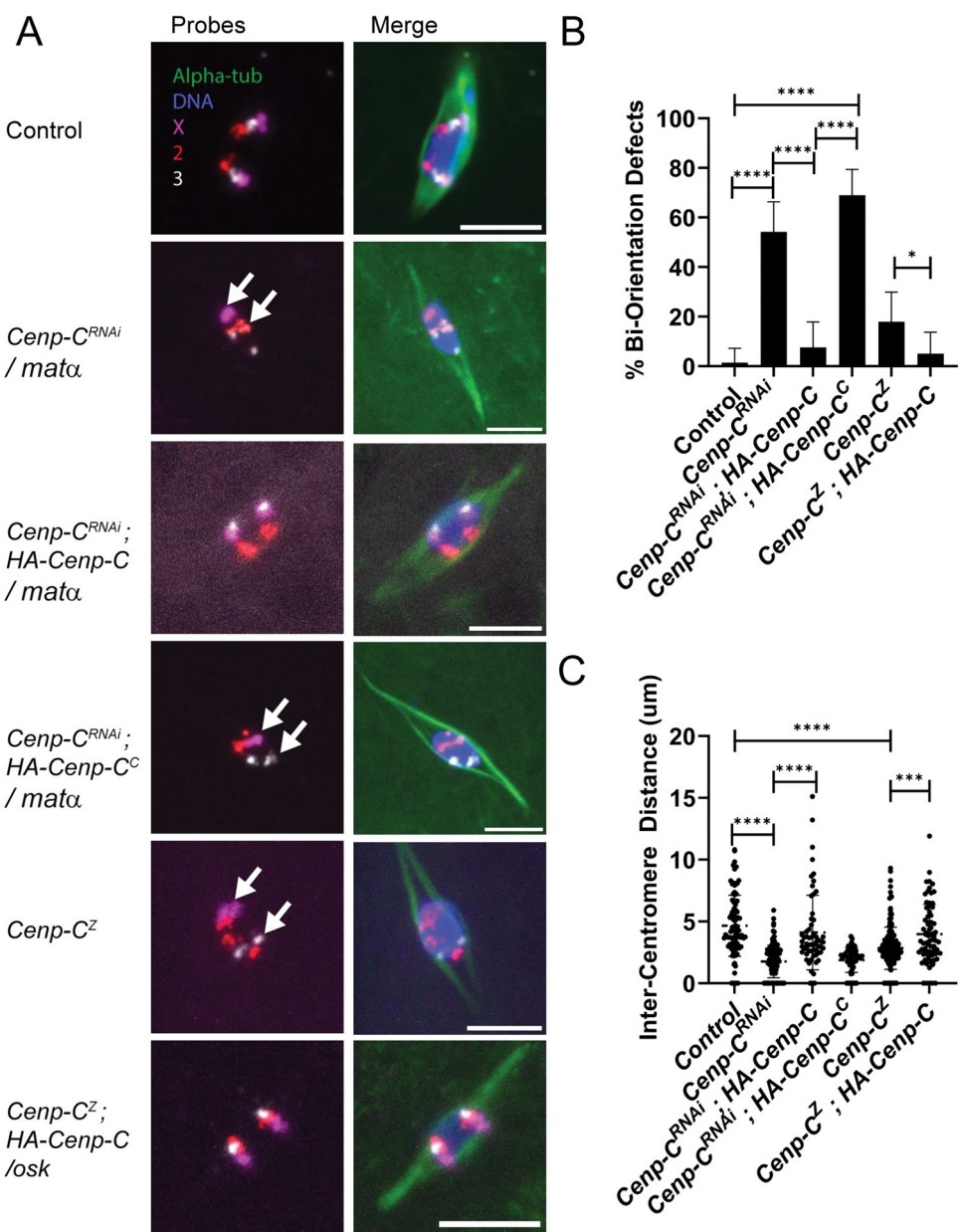

**Fig 6. Prophase-loaded CENP-C is required for bi-orientation in meiosis I.** A) FISH probes were used to detect the pericentromeric regions of chromosome 2 (red), 3 (grey), and X (magenta). Stage 14 oocytes were $Cenp-C^{Z3-4375}/Cenp-C^{IR35}$ ($Cenp-C^Z$) or $GL00409/mata$ ($Cenp-C^{RNAi}$). In some cases, $HA-Cenp-C$ was expressed using $mata$ in $Cenp-C^{RNAi}$ experiments or $oskGal4$ in $Cenp-C^Z$ experiments. B) Bi-orientation defects were defined as two foci of the same probe that were on the same side of the spindle midzone or only a single focus, indicating mono-orientation (n = 75, 59, 53, 58, 56, 60). C) The distance between the center mass of two foci of the same probe, which measures the separation of homologous chromosomes towards opposite poles (n = 90, 156, 60, 60, 192, 72). All scale bars represent 5 μm and error bars represent standard deviation from the mean. * = 0.0473>p>0.0387, **p = 0.0024, *** = 0.0002 >p> 0.001, ****p<0.001.

to a null allele ($Cenp-C^{IR35}$), the females were sterile and centromeric CENP-C levels in oocytes were reduced, though not to the same levels as with $GL00409$ (Fig 5A and 5B). Bi-orientation errors were also elevated in $Cenp-C^{Z3-4375}/Cenp-C^{IR35}$ mutant oocytes, though not to the same extent as the shRNA-expressing oocytes (Fig 6). Expression of HA-CENP-C or GFP-CENP-C

**Table 2. Fertility in *Cenp-C* RNAi, mutant and transgenic females.**

| Genotype | Offspring per female (# of females) |
|---|---|
| Control (*w-*) | 22.9 (90) |
| *HA-Cenp-C; matα* | 0.6 (325) |
| *HA-Cenp-C; oskGal4* | 23.2 (125) |
| *GFP-Cenp-C; matα* | 0.1 (150) |
| *GFP-Cenp-C; oskGal4* | 18.1 (65) |
| *Cenp-C$^Z$/+* | 11.5 (120) |
| *Cenp-C$^{IR35}$/+* | 13.6 (160) |
| *Cenp-C$^{pr141}$/+* | 19.8 (105) |
| *Cenp-C$^Z$/Cenp-C$^Z$* | 1.5 (120) |
| *Cenp-C$^Z$/ Cenp-C$^{IR35}$* | 0.1 (270) |
| *Cenp-C$^Z$/ Cenp-C$^{pr141}$* | 0.2 (70) |
| *Cenp-C$^Z$/ Cenp-C$^{IR35}$; HA-Cenp-C; matα* | 1.4 (460) |
| *Cenp-C$^Z$/ Cenp-C$^{IR35}$; HA-Cenp-C; oskGal4* | 15.2 (180) |
| *Cenp-C$^Z$/ Cenp-C$^{pr141}$; GFP-Cenp-C; matα* | 0.2 (125) |
| *Cenp-C$^Z$/ Cenp-C$^{pr141}$; GFP-Cenp-C; oskGal4* | 11.3 (130) |
| *GL00409; matα* | 0 (120) |
| *GL00409; oskGal4* | 10.8 (90) |
| *GL00409; HA-Cenp-C; matα* | 0.1 (430) |
| *GL00409; HA-Cenp-C; oskGal4* | 26.7 (65) |

using *oskGal4* was found to rescue the sterility of *Cenp-C$^{Z3-4375}$/Cenp-C$^{IR35}$* mutant females (Table 2). In addition, Spc105R localization was restored (Fig 5F–5G) and the biorientation defects in *Cenp-C$^{Z3-4375}$/Cenp-C$^{IR35}$* mutant females was rescued (Fig 6). These results show that the meiosis I and fertility phenotypes of *Cenp-C$^{Z3-4375}$/Cenp-C$^{IR35}$* are likely due to a defect in the prophase I pool of CENP-C.

## Regulation of CENP-C loading and centromere integrity

The RNAi resistance of the *HA-Cenp-C* transgene is a useful tool for expressing mutant variants in the absence of wild-type CENP-C. Previous studies have shown that the C-terminal domain of *Drosophila* CENP-C interacts with the centromere components CID and CAL1 [19,33]. To determine what domains of CENP-C are required for prophase loading in oocytes, we generated two *Cenp-C* transgenes expressing either the N-terminal (aa 1–788, *Cenp-C$^N$*) or C-terminal (789–1411, *Cenp-C$^C$*) domain of CENP-C. The N-terminal domain was not detected by fluorescence microscopy, suggesting it is either unstable or does not localize. In contrast, CENP-C$^C$ was loaded to the centromeres during prophase (S4 and S6 Figs), but failed to recruit Spc105R or Mis12 (Fig 5D–5G). The CENP-C$^C$ protein may have a dominant negative effect because there was a significant reduction of Spc105R localization in *GL00409; Cenp-C$^C$* oocytes compared to *GL00409*. These results support the conclusion that the C-terminal domain of CENP-C is required for centromere localization, while the N-terminal domain recruits outer kinetochore proteins like Mis12 and Spc105R.

Two pathways have been proposed to load CENP-C via the C-terminal domain. The first occurs during anaphase or G1, is required to maintain centromere identity [10,29], and depends on CAL1 [11]. The second occurs during prophase, may be required to build kinetochores, and depends on CDK1 in mitotic cells [65]. To examine the role of CAL1 in prophase loading, we measured CENP-C loading in *cal1* RNAi oocytes (*GL01832*). Similarly, the same experiment was done using an shRNA (*HMS01531*) to knock down *Cdk1* mRNA. In both

experiments, HA-CENP-C was observed in stage 5 oocytes, suggesting neither CAL1 nor CDK1 are required for early prophase loading of CENP-C (S7 Fig). We did not examine stage 14 oocytes because, as described above, CAL1 is not present, and *Cdk1* RNAi oocytes fail to enter prometaphase.

The *Cenp-C*^Z3-4375 mutation is in the C-terminal domain and the protein is present at the centromeres in decreased amounts (Fig 5A and 5B). We also observed a second phenotype suggesting a loss of kinetochore integrity. In some *Cenp-C*^Z3-4375 mutant oocytes, foci of CENP-C protein were detached from the main chromosome mass and moved towards the poles (Fig 7A and 7B). This suggests that one of the defects in *Cenp-C*^Z3-4375 mutant oocytes is a failure to maintain attachment of the kinetochore to the underlying centromeric chromatin. Surprisingly, this phenotype was observed in the presence of a wild-type transgene (Fig 7C). Either the kinetochore phenotype is dominant, or it is due to an early prophase function, and that expression of the transgene, which is in late prophase, was too late to rescue the phenotype.

The kinetochore detachment phenotype is reminiscent of results from budding yeast indicating condensins have a role in maintaining the cohesive structure of the centromeres [66,67]. To compare the two phenotypes, we examined centromere and kinetochore protein localization in oocytes depleted of Condensin subunits. Depletion of SMC2 results in severe loss of centromere integrity (20/22 oocytes, Fig 7E). In contrast to *Cenp-C*^Z3-4375 mutant oocytes, CID was stretched towards the poles in addition to CENP-C. A similar, although less severe, phenotype was observed in *gluon* (SMC1) RNAi oocytes (5/11 oocytes), which could be due to a weaker knockdown. RNAi against subunits specific for Condensin I (Barren/Cap-H, 0/11) or Condensin II (CapH2, 0/6) did not have this phenotype, although we can't rule out the possibility that the absence of a phenotype was due to insufficient mRNA knockdown (Fig 7F). Spc105R was also stretched towards the poles (Fig 7G), and a similar phenotype was observed using *Top2* RNAi (7/7, Fig 7H)), as observed previously [68]. These results show the importance of two mechanisms that resist the pulling forces on the kinetochores and keep the centromere and kinetochore together: Condensin /Top II maintains the compact centromeric chromatin while the C-terminal domain of CENP-C maintains the connection between centromeric chromatin and the inner kinetochore.

## Discussion

### Prophase dynamics of centromere proteins in oocytes

In *Drosophila*, centromere assembly involves three proteins, CID, CAL1, and CENP-C, and in several somatic cell types occurs during late M or G1 [11,18,20,22]. We show here that CENP-C must be loaded during meiotic prophase for centromere clustering, cohesion, and normal kinetochore assembly and this process appears to be independent of CAL1 (Fig 8). We observed increased CENP-C unloading when there was a cytoplasmic source of CENP-C for replacement. This indicates that CENP-C dynamics involves an exchange mechanism, and without new sources of CENP-C, the dynamics are reduced. In contrast to CENP-C, expression of CID in premeiotic cells or early prophase (using *NGTA*) was sufficient for robust localization in metaphase I. Surprisingly, CAL1 is unloaded during prophase and absent in metaphase I (Fig 8), which is in contrast to observations in mitotic cells [9,18] but has been observed in male meiotic prophase [28,29].

The stable property of CID is consistent with studies on CENP-A in mouse oocytes [39]. It has been reported, however, based on changes in fluorescence intensity, that CID is incorporated during meiotic prophase of *Drosophila* sperm and oocytes [24,28,29] (Fig 8). For two reasons, these results do not conflict with ours. First, the two experiments are different; we tested

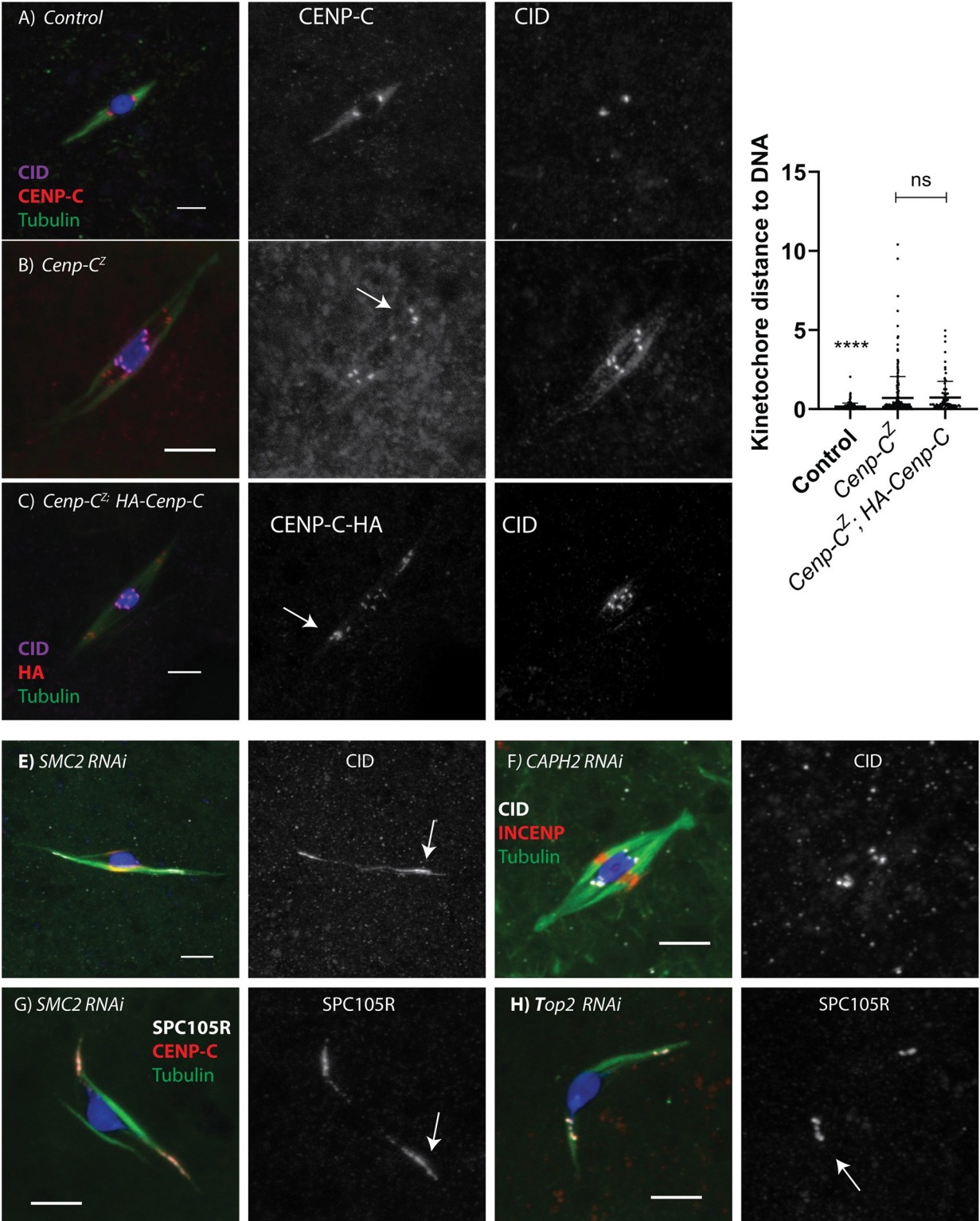

**Fig 7. Kinetochore detachment in *Cenp-C^{Z4.4375}* mutant oocytes.** A-C) Kinetochore detachment in oocytes, with CENP-C in red, CID in magenta, microtubules in green, and DNA in blue. CENP-C foci have separated from the chromosomes (examples shown with arrows), while the CID remain**s** chromatin associated. Shown are (A) wild-type control, (B) *Cenp-C^{Z3.4375} / Cenp-C^{IR35}* (labelled *Cenp-C^Z*), and (C) *HA-Cenp-C* transgene expressed in *Cenp-C^{Z3.4375} / Cenp-C^{pr141}* mutant oocytes, with HA-CENP-C in red. D) The distance between the center of the kinetochore focus and the nearest surface of the DNA was measured. All scale bars represent 5 μm and error bars represent standard deviation

from the mean. (centromere n = 164, 284, 96). **** = p<0.0001. D-E) Requirement of Condensin for kinetochore integrity. *CAPH2* or *SMC2* RNAi oocytes with CID (white), INCENP (red), tubulin (green), and DNA (blue). F-G) *SMC2* or *Top2* RNAi oocytes with Spc105R (white), CENP-C (red), tubulin (green), and DNA (blue). Scale bars represent 5 μm.

if early prophase loading (based on *NGTA*) was sufficient for meiosis centromere localization. The previous experiments tested if CID loads during prophase. Second, we observed small decreases in CID levels when CENP-C was depleted in meiotic prophase. This could be explained by a loss of CID loading, although it is also possible that existing CID is not maintained because CENP-C loading is lost. Thus, it appears that significant amounts of stable CID are loaded prior to meiotic prophase, similar to cohesins [41,56]. However, more is added or exchanged during prophase, which has also been proposed with *Drosophila* cohesins [69]. The function or mechanism of CID prophase loading is not known. Centromeric CAL1 was observed up to stage 5 oocytes, which could be sufficient to load CID during prophase. The significance of CID prophase loading is not known, but an interesting idea is that it prepares centromeres for meiosis-specific kinetochore functions [70].

Over expression of CID during meiotic prophase resulted in ectopic localization to most of the chromatin, recruitment of kinetochore proteins like CENP-C and Spc105R, and sterility. Similar observations of ectopic localization have been made when CID is overexpressed in somatic cells [71–73]. Thus, it may be important to limit CID expression during prophase to avoid localization to non-centromeric locations. Interestingly, the ectopic CID was capable of

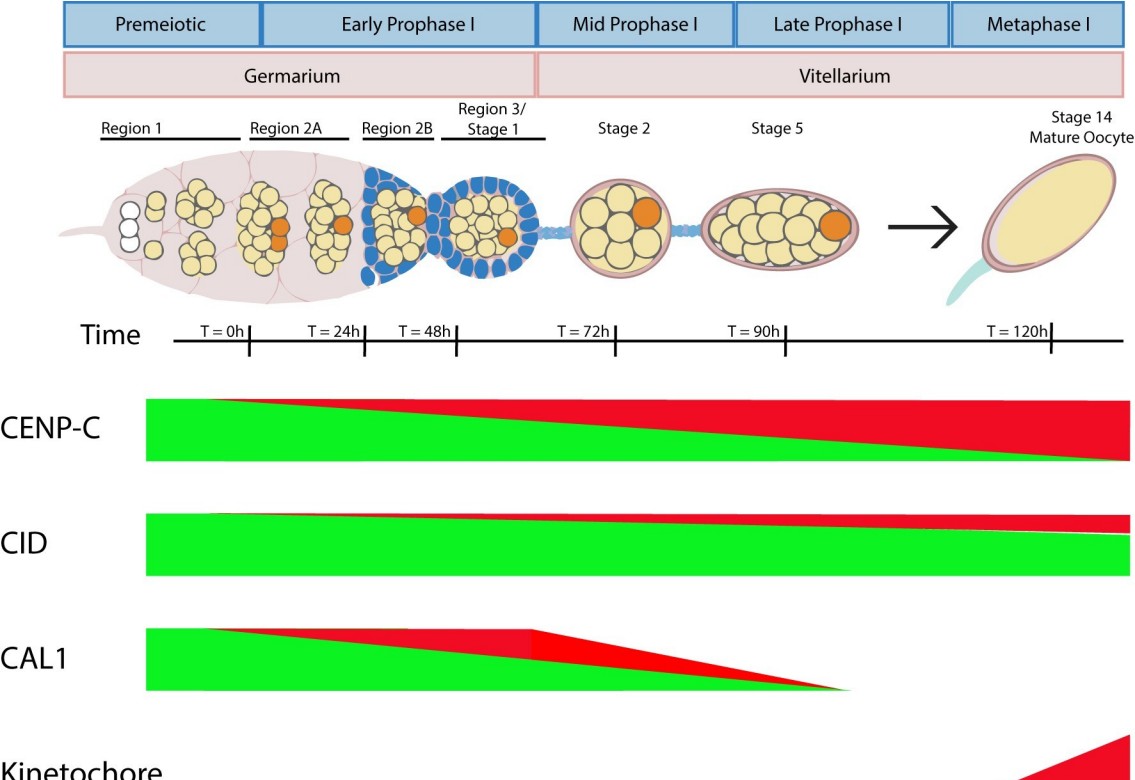

**Fig 8. Model for centromere and kinetochore dynamics in prophase.** The bars show relative levels of expression, with green being expression in the germarium (premeiotic pool) and red is prophase expression (prophase I pool). CENP-C is compared to a representation of the data from the literature (see Discussion for details). While a prophase I pool of CID has been proposed [29], it is possible there is no CAL1 prophase expression.

recruiting CENP-C and kinetochore proteins, suggesting there is a CENP-C independent mechanism for CID recruitment.

The C-terminal domain of CENP-C had similar dynamic properties during prophase as the full-length protein. This part of the protein also contains the elements required to interact with CID and CAL1 for centromere maintenance [19,20,35]. Thus, CENP-C prophase loading may involve recruitment to existing sites containing CID. Interestingly, Kwenda et al. [24] showed that *cal1* mutant oocytes failed to load CID during prophase, but still showed loading of CENP-C, indicating that CENP-C loading is at least partially independent of CID. Some evidence in mammals, however, suggests CENP-C has both CENP-A-dependent and independent localization mechanisms, although in most of these studies, loading during G2 is not specifically addressed [74]. CDK1 has been implicated in CENP-C dynamics in chicken and human cells [65], although we did not see an effect on CENP-C prophase loading when CDK1 was depleted in *Drosophila* oocytes. Further work is needed to understand the mechanism of CENP-C dynamics during meiotic prophase.

## Early prophase function: centromere clustering and cohesion

When CENP-C was depleted from early prophase oocytes, we observed an increased number of centromere foci. Our results suggest the increased CID foci phenotype could be due to a loss of centromeric cohesion because we also observed an increase in sister chromatid nondisjunction. These observations are consistent with and extend prior studies on *Cenp-C* [58]. *Cenp-C*^Z3-4375^ mutant females have defects in centromere clustering in pachytene oocytes (region 3 of the germarium) and decreased localization of the cohesion protection protein MEI-S332 [24]. Our results suggest that the prophase I pool of CENP-C is required to recruit or maintain cohesion. A recruitment function of CENP-C could occur when cohesion is established in S-phase [56]. Alternatively, our results show striking similarities to the known roles of mammalian and yeast CENP-C in recruiting Moa1 or Meikin [75,76], which are protectors of centromeric cohesion and particularly important for meiosis I.

## Late prophase function: kinetochore assembly

Depletion of the prophase pool of CENP-C resulted in phenotypes associated with loss of the kinetochore, including the loss of Mis12 and Spc105R. These results show that CENP-C is required for most or all kinetochore assembly, consistent with studies in *Drosophila* mitotic cells [22,34,35]. These results also show that unlike CID/CENP-A, CENP-C loaded prior to prophase, that is required for centromere maintenance, is not sufficient for kinetochore assembly. Why CENP-C is loaded during prophase instead of prometaphase I like most other kinetochore proteins is not known. Studies in vertebrate cells have shown that CENP-C is dynamic during interphase but stable in metaphase [43,77]. In addition, CENP-C turnover is much slower compared to microtubules and kinetochore proteins [78]. Therefore, it may be necessary to assemble CENP-C at the centromeres during prophase because its dynamics are much slower than other spindle proteins. Prophase loading may also provide a stable and defined platform for kinetochore assembly at the beginning of meiotic prometaphase in oocytes.

The C-terminal domain of CENP-C was recruited to the centromeres but failed to assemble a kinetochore. Thus, the N-terminal domain of CENP-C appears to interact with Mis12 and recruit the rest of the kinetochore, consistent with previous studies in *Drosophila* [33,38]. The *Cencp-C*^Z3-4375^ mutation, which causes a substitution of amino acid 1115 [58], appears to weaken the interaction between CENP-C and the centromere. This is likely the cause of the detached kinetochore phenotype in this mutant, which indicates a failure to maintain kinetochore integrity when experiencing the force of microtubule pulling. We have also shown that

Condensin, along with Topoisomerase II [68], is required to maintain the integrity of CENP-C and other kinetochore proteins on the chromosomes. Previous work in yeast has suggested that condensins have a role in maintaining the cohesive structure of the centromeres in metaphase [66,67].

## Implications of the CENP-C prophase I pool

CENP-A/CID is remarkably stable [79,80], and the overexpression of CID results suggests that limiting CID loading in prophase may be necessary to maintain a single centromere on each chromosome. In contrast, CENP-C must be loaded during prophase for cohesion maintenance and kinetochore assembly. Similarly, CENP-C is loaded in prophase I oocytes of the mouse [39]. Why CENP-C is not like CENP-A/CID and must be loaded during prophase is not known. It is possible that more CENP-C is needed for kinetochore assembly than centromere maintenance. It may simply be that, for unknown reasons, CENP-C is not as stable as CENP-A, and therefore requires replenishment during the prophase pause to maintain functional levels. Regardless of the reason, the requirement for CENP-C prophase loading could affect the health of aging oocytes. If loading of CENP-C is compromised in oocytes that have spent a longer time in meiotic prophase, our results show that sister chromatid cohesion, kinetochore assembly, and meiotic chromosome segregation could be defective, leading to higher levels of aneuploidy.

## Materials and methods

### *Drosophila* strains and genetics

*Drosophila* crosses and stocks were kept at 25°C on standard medium. Fly stocks, including lines expressing a short hairpin RNA (shRNA), were obtained from the Bloomington Stock Center or the Transgenic RNAi Project at Harvard Medical School [TRiP, Boston, MA, USA, flyrnai.org]. Information about the genetic loci can be found on FlyBase [flybase.org]. *Drosophila* were obtained from the Bloomington Stock Center. Expression of the shRNA was controlled by the Upstream Activating Sequence (UAS), which is activated by expression of GAL4 under the control of a tissue-specific enhancer [1]. Five *Gal4* strains were used in this study, four of which were germline-specific: *P{matalpha4-GAL-VP16}V37* and *P{w[+mC] = osk-GAL4::VP16}A11* promote expression after zygotene, *P{GAL4-nos.NGT}A* promotes expression only in the germarium (during prophase I), and *P{GAL4:: VP16-nos.UTR}CG6325^{MVD1}* promotes expression throughout the whole germline. In addition, heat shock was used to induce expression of transgenes using *P{GAL4-Hsp70,PB}89-2-1*.

Two shRNAs were used to reduce *Cenp-C* expression, *GL00409*, located on chromosome 2, and *HMS01171*, located on chromosome 3. To measure the extent of the *Cenp-C mRNA* knockdown, total RNA was extracted from stage 14 oocytes expressing *GL00409* with *P{matalpha4-GAL-VP16}V37* using the TRIzol Reagent (Life Technologies). cDNA was made using the High-Capacity cDNA Reverse Transcription kit (Applied Biosystems). TaqMan Gene Expression Assays (Life Technologies) were used to measure expression levels of CENP-C. RT-qPCR was done on a StepOnePlus (Life Technologies) real-time PCR system and results showed that the *Cenp-C* RNAi expressed only 4% of wild-type RNA. In addition, CAL1 was knocked down using shRNA *GL01832* and CDK1 was knocked down using shRNA *HMS01531*, both located on chromosome 2. Both shRNAs were sterile in combination with *P{matalpha4-GAL-VP16}V37* and *HMS01531* oocytes failed to enter meiosis I.

*Cencp-C^{Z3-4375}* is a homozygous-viable allele that has a missense mutation in the CENP-C motif, changing a proline amino acid to a serine at position 1116 [58]. For the analysis of *Cenp-C^{Z3-4375}*, trans heterozygous mutant strains were used, with one chromosome carrying either the

*Cenp-C$^{IR35}$* or *Cenp-C$^{pr141}$* mutation. The *Cenp-C$^{IR35}$* mutation is a premature stop codon at position 858 before the CENP-C motif and Cupin domain, resulting in a deletion of these regions [81]. The *Cenp-C$^{pr141}$* mutation is a premature stop codon at position 1107 within the CENP-C motif [81], resulting in a partial deletion of the CENP-C motif and the rest of the protein.

### HA- and GFP-tagged transgenes

The coding region of cDNA clone FI18815 was PCR amplified and cloned in frame into pENTR4 (HiFi assembly, NEB) and then into pPHW using Clonase (Life Tech.). The target sequence for *GL00409* is located in the 5'UTR of *Cenp-C* and, therefore, this transgene is RNAi resistant. Mutant forms of *HA-Cenp-C* were constructed using the Q5 Site-Directed Mutagenesis Kit (NEB). The GFP-CENP-C transgenes were constructed by Christian Lehner and contain the 5'UTR sequences upstream of the GFP sequence, and thus are RNAi sensitive. Mis12 localization was observed using a UASP regulated GFP-fusion transgene [82].

### Fertility trials, nondisjunction, and crossover assays

The *GL00409 Cenp-C* shRNA was tested for fertility and chromosome segregation errors by crossing females expressing the shRNA to *yw/ B$^S$ Y* males. The aneuploid genotypes that survive are *y/y/ B$^S$ Y* (Bar-eyed females) and *y w / 0* (wild-type males). Because half the aneuploid progeny have lethal genotypes, aneuploidy was calculated by multiplying the number of aneuploid progeny by two and dividing by the total number of progeny.

Sister chromatid aneuploidy was tested by crossing *y w/ Bwinscy* females expressing the shRNA to *v f B ^ Y* males. If aneuploidy occurs among homologous chromosomes, the expected genotypes that survive are *y w/Bwinscy* and *0/ v f B ^ Y*. Aneuploidy among sister chromatids is detected by the genotypes *y w/y w* (non-bar-eyed females) or *Bwinscy/Bwinscy* (Bar-eyed females). Because the *0/ v f B ^ Y* genotype arises from both MI or MII nondisjunction, only the *y w/y w* and *Bwinscy/Bwinscy* progeny were used to determine the sister chromatid aneuploidy frequency. Therefore, the number of aneuploid progeny counted was multiplied by four and divided by the total number of progeny in order to measure sister chromatid aneuploidy.

In order to observe the role of CENP-C in crossing over, females were generated that expressed the shRNA and were heterozygous for four genetic markers on chromosome III: *st*, *cu*, *e* and *ca*. These females were crossed to a strain carrying all the recessive traits and the frequency of the recombinants was scored.

### Cytology and immunofluorescence of early prophase oocytes

For immunolocalization experiments in the germaria, mated females were aged for 1–2 days at 25˚C. In one well of a two well plate, 10–15 ovaries were dissected using 1x Robb's media and then were transferred to the second well containing fresh media. A tungsten needle was used to break open the ovary sheath and tease the ovarioles apart. The dissected ovaries were transferred to a 1.5 mL Eppendorf tube with 4% formaldehyde in 500 ul of Buffer A as described [83]. The ovaries were nutated at room temperature for 10 minutes, then were washed four times before adding the primary antibodies. The next day, following four washes, secondary antibodies were added and incubated at room temperature for 4 hours.

### Cytology and immunofluorescence of metaphase I oocytes

For immunolocalization experiments in pro-metaphase I oocytes (stages 13–14), we used the immunocytochemical protocol as described [84]. In brief, 100–200 females were aged 2–3 days

with males in yeasted vials [85]. Oocytes were collected by pulsing the females in a blender and then separating the oocytes from the bulk fly tissues using a mesh. Oocytes were fixed in 5% formaldehyde solution for 2.5 minutes, and then equal amounts of heptane were added and the oocytes were vortexed for 30 seconds. The membranes were removed by rolling the oocytes between a coverslip and the frosted part of a glass slide. These oocytes were incubated in PBS/ 1% Triton X-100 for two hours, then washed in PBS/0.05% Triton X-100. The oocytes were blocked in PBS/0.1% Tween 20/0.5% BSA (PTB) for one hour, and then incubated with primary antibodies overnight. Oocytes were washed the next day in PTB and incubated with secondary antibodies for 4 hours at room temperature.

Tissues were mounted for confocal imaging using SlowFade Gold (Invitrogen). A Leica TPS SP8 confocal microscope with a 63X, N.A. 1.4 lens was used to visualize fluorescent tags using different colored lasers. Images were imaged by collecting sections throughout the germarium or stage 14 oocyte spindle, using parameters optimized by the Leica Confocal software based on the lens and wavelength. Images were analyzed as image stacks and presented as maximum projections of whole germarium, cells, or spindles.

## Antibodies

An antibody against CENP-C made in guinea pig was made by generating a clone expressing amino acids 502–939 (Genscript). This guinea pig anti-CENP-C was used at 1:1000. Additional primary antibodies were rat anti-CID (Active motif, 1:100), rabbit anti-CID (Active motif, 1:100), rabbit anti-Spc105R (1:4000) [86], rabbit anti-GFP (Invitrogen, 1:400), rat anti-HA (Roche, 1:50), mouse anti-C(3)G (1:500) [87], two mouse anti-ORB antibodies, 6H4 and 4H8 (1:100 for each) [88] and mouse anti-α tubulin DM1A conjugated directly to FITC (Sigma, 1:50). The secondary antibodies that were used were Cy3, Alexa 546, Alexa 633, or Alexa 647 from Jackson Immunoresearch Laboratories, and Alexa 488 from Invitrogen. The oocytes were stained with Hoechst 33342 at 1:10,000 (10 μg/ml). FISH probes were obtained from IDT where the X359 repeat was labeled with Alexa 594, the dodeca repeat was labeled with Cy5, and the AACAC repeat was labeled with Cy3.

## Quantification and statistical analysis

Aneuploidy in flies expressing the *Cenp-C* shRNA with NGTA was compared to that of the control group and an unpaired t-test was done to determine if the difference is statistically significant (p-value < 0.05). Sister chromatid aneuploidy in flies expressing the *Cenp-C* shRNA with NGTA was compared to that of the control group and a t-test was used to determine whether the difference was statistically significant (p-value < 0.05). The percent of crossovers at each position was compared to that of the control group using a t-test, and if the difference was statistically significant (p-value < 0.05), then this indicated that elevated or reduced recombination occurred at that specific location.

Quantification of CID foci and protein localization in both the germarium and in stage 14 oocytes was measured using the Imaris Software. To quantify centromere foci, the automated spots detection feature of Imaris was used. A spot with a XY diameter of 0.20 μm, a Z diameter of 1.00 μm, and a physical interaction with the DNA was counted as a centromere. A t-test was done to compare the number of centromere foci in flies expressing the knockdown to the number in the control group using a p-value < 0.05 to indicate statistical significance.

To measure the localization of proteins to the centromere or the nucleus in *Cenp-C* RNAi, intensity experiments were performed using Imaris. The Imaris software was utilized to measure protein intensity at the centromere of the oocyte, using CID as the centromere marker. Spots were chosen in random somatic cells to quantify the intensity of the

background. The intensity of the protein at the centromere was divided by the background and a t-test was used to determine whether the centromere protein intensity was reduced with a loss of CENP-C.

Intensity experiments were also performed to measure localization of Mis12, Spc105R, or CENP-C to the centromere at stage 14 using a similar protocol. However, background intensity was measured in the oocyte cytoplasm as opposed to measuring the background in the somatic cells for germarium images. A t-test was used to determine whether the intensity of the protein of interest at the centromere normalized to the background was reduced with a loss of CENP-C using a p-value $<0.05$ to indicate statistical significance.

## Supporting information

**S1 Fig. GFP-CENP-C, CID-GFP or CAL1-GFP in early prophase oocytes.** A) Localization of GFP-CID (green) and centromeres detected using a CENP-C antibody (red). B) Localization of GFP-CAL1 (green) with the centromeres detected using a CID antibody (red). C) Localization of GFP-CENP-C, with the centromeres detected using a CID antibody (red). In all images, the DNA is blue and the scale bars are 5 μm. D) Localization of GFP-CAL1 (green) using *matα* in early stages of the vitellarium (stages 1–5). GFP-CAL1 protein can been seen accumulating in nurse cell nuclei (arrows), but not at the centromeres in the oocyte. The centromeres were detected using a CENP-C antibody (red) and the cytoplasmic ORB protein (white) is enriched in the oocyte. The scale bar is 10 μm.
(TIF)

**S2 Fig. GFP-CENP-C, CID-GFP or CAL1-GFP in stage 14 oocytes.** The DNA is blue, microtubules in white, and the scale bars represent 5 μm. **A)** Localization of CID-GFP or CAL1-GFP (green) using the *MVD1* or *matα*. In the one image (*gcal1-GFP)*, CAL1-GFP is regulated by the endogenous *cal1* promoter. The centromeres were detected using CENP-C (red). **B)** Localization of CID-GFP or GFP-CENP-C (green) using *matα*. The kinetochores were detected using an antibody against Spc105R (red).
(TIF)

**S3 Fig. Expression pattern of *NGTA*.** The expression pattern of **(A)** *P{GAL4-nos.NGT}* and **(B)** *P{GAL4::VP16-nos.UTR}CG6325MVD1* using *UASP-β-galactosidase* as a reporter. Arrowheads indicate anterior tip of the ovariole, where the germarium is located, and the blue stain indicates where each GAL4 promotes expression.
(TIF)

**S4 Fig. Loading of HA-CENP-C during oocyte meiotic prophase.** In all images, HA-CENP-C is green, the centromeres are marked with CID (red), and DNA is in blue. The scale bars represent 5 μm. **A)** Whole germarium with HA-tagged CENP-C expressed using *NGTA* or *matα*. ORB (white) is enriched in the oocyte. **B)** HA-tagged CENP-C or CENP-C[C] was expressed using *matα*. **C)** HA-CENP-C was expressed using *hsp70-Gal4*. Oocytes were collected and fixed after 6 hours (early prophase) or 5 hours (stage 14) after a 1-hour incubation at 37˚C.
(TIF)

**S5 Fig. SC assembly when CENP-C is depleted in prophase.** Confocal images of the germarium with *Cenp-C* RNAi (*HMS01171*) with (A) no *GAL4* and (B) *NGTA*. DNA is shown in blue, CENP-C is in red, and C(3)G is in green. The scale bar is 10 μm. CENP-C and C(3)G are shown in white in the single channel images. Region 1 of the germarium has been boxed to show increased centromeric C(3)G. The insets show single nuclei from region 1 in the

germarium to show co-localization of CENP-C and C(3)G (Scale bar = 3 μm).
(TIF)

**S6 Fig. Additional images of CENP-C localization in RNAi and transgenic oocytes. A)** *Cenp-C* RNAi or *Cenp-C^Z* oocytes with CENP-C (green) and CID (red). **B)** *Cenp-C* RNAi or *Cenp-C^Z* oocytes expressing a *Cenp-C* transgene, with HA in red and Spc105R in green, DNA in blue, microtubules in white, and the scale bars represent 5 μm. **C)** Oocytes shown in panels A and B were assessed for KT-MT attachments. This was done by measuring the distance between each centromere and the nearest microtubule. **D)** Number of centromere foci was measured based on CID foci in *Cenp-C* RNAi metaphase I oocytes (n = 19 and 36). Error bars represent standard deviation.
(TIF)

**S7 Fig. Loading during prophase does not depend on CDK1 or CAL1.** Localization of HA-CENP-C in stage 4–5 oocytes of **(A)** *cdk1* and **(B)** *cal1* RNAi oocytes. HA is in red, CID is in white, cytoplasmic ORB protein is in green, and DNA is in blue. The scale bar represents 5 μm. **C)** Relative intensity of CENP-C in control and RNAi oocytes (n = 18, 21, 18).
(TIF)

## Acknowledgments

We thank Marina Druzhinina for technical assistance, Christian Lehner for providing Drosophila stocks and antibodies. We also thank Sarah Radford for early work on this project. We thank the Bloomington *Drosophila* Stock Center (NIH P40OD018537) and the TRiP project at Harvard Medical School for providing fly stocks used in this study.

## Author Contributions

**Conceptualization:** Jessica E. Fellmeth, Kim S. McKim.

**Formal analysis:** Jessica E. Fellmeth, Hannah Sturm, Kim S. McKim.

**Funding acquisition:** Kim S. McKim.

**Investigation:** Jessica E. Fellmeth, Janet K. Jang, Manisha Persaud, Hannah Sturm, Neha Changela, Aashka Parikh.

**Methodology:** Jessica E. Fellmeth.

**Project administration:** Kim S. McKim.

**Supervision:** Jessica E. Fellmeth, Kim S. McKim.

**Visualization:** Jessica E. Fellmeth, Janet K. Jang, Manisha Persaud, Hannah Sturm.

**Writing – original draft:** Jessica E. Fellmeth, Kim S. McKim.

**Writing – review & editing:** Jessica E. Fellmeth, Kim S. McKim.

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
