## [Decision Letter · Decision Letter 0]

19 Aug 2023

Dear Kim,

Thank you very much for submitting your Research Article entitled 'A Dynamic population of prophase CENP-C is required for meiotic chromosome segregation' to PLOS Genetics.

The manuscript was fully evaluated at the editorial level and by independent peer reviewers. The reviewers appreciated the attention to an important problem, but raised some substantial concerns about the current manuscript. Based on the reviews, we will not be able to accept this version of the manuscript, but we would be willing to review a much-revised version. We cannot, of course, promise publication at that time.

As you will read in the reviewers' comments major concerns were raised regarding the clarity of the paper, appropriate citations, and alternative interpretations to the data presented in the manuscript. Both reviewers commented that the Introduction was lacking in depth and appropriate citations for previous work, thus not providing sufficient context and background for your results. This also applies to the Discussion (albeit to a lesser extent) and at least one section of the Results (Line 198-199) where previous work was not explicitly acknowledged and cited. Please take care to correct this in a revised manuscript. Both reviewers commented on the quality of the figures, in addition to details that should be revised to make them more readable. Please take care to submit higher resolution figures. A major concern is that the GFP-tagged proteins are not expressed under their normal regulatory sequences. Although this provides temporal control, it also is a limitation that needs to be more carefully considered. If the genetics backgrounds have endogenous protein expression, then does this may constitute over expression conditions-- Reviewer #1 expressed detailed concerns that this possibility was not properly addressed. Moreover, reviewer #2 details a major concern in point #10 that needs to be addressed with new experiments-- either co-expression of GFP-tagged CID and CAL1 or differentially tagged constructs. Please see the reviewers comments for more details and helpful alternative approaches recommended by the reviewers.

Given that the reviewers comments and concerns are extensive, you may wish to consider sending a revised version of your manuscript to an alternative journal. Should you decide to revise the manuscript for further consideration here, your revisions should address all the specific points made by each reviewer. We will also require a detailed list of your responses to the review comments and a description of the changes you have made in the manuscript.

If you decide to revise the manuscript for further consideration at PLOS Genetics, please aim to resubmit within the next 60 days, unless it will take extra time to address the concerns of the reviewers, in which case we would appreciate an expected resubmission date by email to plosgenetics@plos.org.

We are sorry that we cannot be more positive about your manuscript at this stage. Please do not hesitate to contact us if you have any concerns or questions.

Yours sincerely,

Giovanni Bosco, Ph.D.

Academic Editor

PLOS Genetics

Gregory P. Copenhaver

Editor-in-Chief

PLOS Genetics

Reviewer's Responses to Questions

**Comments to the Authors:**

Reviewer #1: The paper by Fellmeth et. al. addresses the roles of centromeric proteins CENP-C, CENP-A/CID and the loader CAL1 in the Drosophila oogenic gremline. The paper utilized powerful genetic tools to dissect early and late meiotic functions. They discover that early expression of CID is sufficient for its localization to centromeres in oocytes and restricting its meiotic accumulation may be important for suppressing ectopic centromeres. Their studies propose that CENP-C is constantly loaded throughout prophase, with early prophase loading important for centromeric cohesion. Late loading of CENP-C is required for kinetochore protein recruitment. The studies are well done and, generally speaking, clearly presented. I have some suggestions for improving the text and figures in a way that will be accessible to readers outside the first field of study.

• Introduction: key information that is missing her is the localization pattern of CID and CENP-C in Drosophila oogenesis. There is a discussion of their mitotic localization, but there is no clear statement regarding Drosophila oogenesis. This information is already published (e.g., Fellmeth and Kim 2020 Fig 1) and is important for understanding the experiments presented. In addition, it is important to indicate it in the introduction (and not wait for the discussion) that CID progressively assembles during meiotic prophase (Dunleavy 2012).

• The description of experiments could be improved to facilitate understanding of the experimental design by the general (scientific) public. I suggest the following points explained in more details:

o Line 117-125: I assume that all the experimental conditions in Figure 1 are UASP-driven expression in the background that contains the endogenous genes. It should be stated clearly. The reader should be informed early on that this can lead to overexpression (for all conditions and not just CID).

o Line 176-182: clarify that all endogenous CENP-C is GFP tagged

o Line 201-202: The authors move from describing driving protein to driving shRNA expression and if the reader misses that, then the whole next section is very confusing. “To investigate Cenp-C function in early prophase, we knocked down its expression by RNAi using an shRNA (GL00409) with either the NGTA or MVD1 promoters.” I would expand the description here (and break to 2-3 sentences) to specifically state that Cenp-c is the endogenous copy and that the shRNA driven from MVD1 will result in knockdown of cenp-c throughout the germline and shRNA driven from NGTA will result in knockdown of cenp-c early in meiosis.

• The quality of the graphs is very low on this version. It was very hard to understand them as the axis labeling was frequently pixelated. This must be fixed for publication. Figure 1E should be enlarged. I assume that in publication it won’t take the whole page and therefore will be smaller (and unreadable the way it is now).

• If indeed all experiments in Figure 1 were done in the background of wild type copy of the protein, then these are also over expression conditions. It is not clear why overexpression effect on localization is only considered for CAL1 and therefore only for it a control experiment is performed. Including the localization patten of CID and CENP-C driven by endogenous promoter in Fig 1 will help. The endogenous protein localization pattern should be shown in Figure 1 (for stage 1 and 14), so it can be compared to the stage-specific expression. There is some data about that for CAL1 in S2 but not for earlier stages.

• The organization of figure 1 is confusing and is not in line with the text. The text discuses MVD1 before NGTA but the figure order is reversed. Same is with graph in the figure

• The description of the CID-MVD1 (line 129-131) needs to be rewritten in a way the reflects figure 1E. The way it reads now is as if there are only 2 categories

• Figure 2E- the order on the x axis should be the same between the graphs. The labeling of the X axis should be explained in the figure legend. I understand it’s a shorter version of B-D but it is confusing since its different. the Y axis should be the same in all graphs in E.

• The authors need to explain why loss of cohesion leads to increase in crossover. Do they see more centromeric DSBs or is this a result of a change in CO/NCO decision post-DSB formation?

• Figure 3-5 the RNAi is stage-specific and must be differentiated by introducing different naming for each RNAi condition in the figure panels. Just saying RNAi implies it’s all over the organism.

• Discussion: CID localization is only partially dependent on CENP-C and its overexpression is sufficient for formation of ectopic chromatin-associated foci. Is it possible there is a CENP-C independent mechanism for CID recruitment? This option should be discussed.

The authors show here that CAL1 is unloaded in prophase. Dunleavy 2012 showed that CID levels are increased during female prophase. CAL1 is also a chaperon loader of CID. It is therefore unclear what loads CID in prophase. Can this be discussed?

• Discussion: Page 23, Line 434 the authors cite mammalian papers to say that CENP-A/CID is not loaded in prophase. However, in Drosophila oocytes (Dunleavy 2012, Figure 3B) CID continuously loads. Related- Page 20, line 379: Why do the others state that the loading of CID in prophase is small? I understand the argument that restricted loading is fine but high levels of lodging are detrimental to oogenesis, but unless the data in Dunleavy 2012 is incorrect I don’t think the authors should downplay prophase loading of CID.

• A model describing the findings of this paper and comparing them to mammalian systems should be added.

Reviewer #2: See attachment

**Have all data underlying the figures and results presented in the manuscript been provided?**

Reviewer #1: Yes

Reviewer #2: Yes

PLOS authors have the option to publish the peer review history of their article (what does this mean?). If published, this will include your full peer review and any attached files.

Reviewer #1: No

Reviewer #2: No

---

## [Decision Letter · Decision Letter 1]

14 Nov 2023

Dear Kim,

We are pleased to inform you that your manuscript entitled "A Dynamic population of prophase CENP-C is required for meiotic chromosome segregation" has been editorially accepted for publication in PLOS Genetics. Congratulations!

Yours sincerely,

Giovanni Bosco, Ph.D.

Academic Editor

PLOS Genetics

Gregory P. Copenhaver

Editor-in-Chief

PLOS Genetics

Comments from the reviewers (if applicable):

Reviewer's Responses to Questions

**Comments to the Authors:**

Reviewer #1: My concerns have been addressed

Reviewer #2: The authors have adequately addressed most of my concerns.

**Have all data underlying the figures and results presented in the manuscript been provided?**

Reviewer #1: None

Reviewer #2: Yes

PLOS authors have the option to publish the peer review history of their article (what does this mean?). If published, this will include your full peer review and any attached files.

Reviewer #1: No

Reviewer #2: No

**Data Deposition**

http://datadryad.org/submit?journalID=pgenetics&manu=PGENETICS-D-23-00807R1

**Press Queries**

---

## [Editor Report · Acceptance letter]

23 Nov 2023

PGENETICS-D-23-00807R1 

A Dynamic population of prophase CENP-C is required for meiotic chromosome segregation 

Dear Dr McKim, 

We are pleased to inform you that your manuscript entitled "A Dynamic population of prophase CENP-C is required for meiotic chromosome segregation" has been formally accepted for publication in PLOS Genetics! Your manuscript is now with our production department and you will be notified of the publication date in due course.

With kind regards,

Judit Kozma

PLOS Genetics

On behalf of:
